# Significant decline of mesospheric water vapor at the NDACC site Bern in the period 2007 to 2018

Martin Lainer[1,a], Klemens Hocke[1], Ellen Eckert[2], and Niklaus Kämpfer[1]

[1]Institute of Applied Physics, University of Bern, Bern, Switzerland
[2]University of Toronto, Department of Physics, 60 St. George Street, Toronto, Ontario M5S 1A7, Canada
[a]now at: Swiss Federal Office of Meteorology and Climatology, Locarno-Monti, Switzerland

**Correspondence:** Martin Lainer (martin.lainer@meteoswiss.ch)

**Abstract.** The middle atmospheric water vapor radiometer MIAWARA is located close to Bern in Zimmerwald ($46.88°$ N, $7.46°$ E, $907$ m) and is part of the Network for the Detection of Atmospheric Composition Change (NDACC). Initially built in the year 2002, a major upgrade of the instruments spectrometer allowed to continuously measure middle atmospheric water vapor since April 2007. Thenceforward to May 2018, a time series of more than 11 years has been gathered, that makes a first trend estimate possible. For the trend estimation, a robust multi-linear parametric trend model has been used. The trend model encompasses a linear term, a solar activity tracker, the El Niño–Southern Oscillation (ENSO) index, the quasi-biennial oscillation (QBO) as well as the annual and semi-annual oscillation. In the time period April 2007 to May 2018 we find a significant decline in water vapor by $-0.6 \pm 0.2 \, \mathrm{ppm \, decade^{-1}}$ between 61 and 72 km. Below the stratopause level ($\sim 48$ km) a smaller reduction of $H_2O$ of up to $-0.3 \pm 0.1 \, \mathrm{ppm \, decade^{-1}}$ is detected.

## 1 Introduction

Water vapor is the most important greenhouse gas in the atmosphere (Kiehl and Trenberth, 1997) and has a dominant feedback role in the Earth's climate system. In the troposphere it provides the main source of moisture for the formation process of precipitation in the atmosphere. While global warming progresses, the amount of moisture is expected to increase faster than the overall amount of precipitation, that is controlled by evaporation and the heat budget at the surface (Trenberth et al., 2003).

Long-term changes in the abundance of atmospheric water vapor can be used to characterize climate change. One region of the atmosphere which is very sensitive to those changes is the upper troposphere, but the actual impact on climate change is poorly understood (Held and Soden, 2000). Some direct anthropogenic changes in water vapor are due to emissions by aviation and the possible subsequent formation of contrails that freeze-dry the air and exert a strong radiative forcing (RF) effect. Contrails that persist for several hours and loose their line shaped form are known as contrail-cirrus. Globally averaged (1999 to 2016), annual mean RF estimates with uncertainty ranges are about $0.01$ (0.005-0.03) $\mathrm{W \, m^{-2}}$ for long-lived contrails alone, and together with contrail-cirrus RF reaches about $0.05$ (0.02-0.15) $\mathrm{W \, m^{-2}}$ (Kärcher, 2018). In contrast, total aviation RF for instance in the year 2000 is about $0.048 \, \mathrm{W \, m^{-2}}$ (Sausen et al., 2005).

Compared to the troposphere, the stratosphere is very dry and the amount of $H_2O$ is commonly indicated in volume mixing ratios (parts per million) like for ozone. Water vapor from the troposphere can enter the stratosphere mainly through convective

processes at the equator. The cold tropical tropopause acts as a cold trap for ascending tropospheric air and causes most of the water vapor to freeze out. Nevertheless, water vapor in the stratosphere has a high impact on ozone chemistry and it is of importance to a global warming feedback process. Further, water vapor provides the main source of hydrogen radicals (OH, H, $HO_2$), which are involved in the catalytic destruction cycle of ozone in the stratosphere (Brasseur and Solomon, 2006). An important long-term data set of lower free tropospheric ($2\,km$) up to middle stratospheric ($28\,km$) water vapor is available from Boulder (Colorado) since 1980. This data comes from balloon frost-point hygrometer (FPH) measurements that are launched usually once per month. A weighted, piecewise regression analysis of the 30-year record from 1980 to 2010 by Hurst et al. (2011) revealed an average increase by $1.0 \pm 0.2\,ppm$ in the altitude range between 16 and $26\,km$. About a quarter of the $H_2O$ increase could be attributed to changes in the methane ($CH_4$) concentration. Methane can easily be transported from the surface upward into the stratosphere where its oxidation is a major in-situ source of water vapor.

Compared to water vapor, stratospheric ozone gathered much higher scientific attention in regard of its long-term development after the detection of the Antarctic ozone whole in 1985 (Farman et al., 1985). Two years later in 1987 the Montreal Protocol has been signed to protect the ozone layer by banning and regulating the production of numerous substances that are responsible for ozone depletion. Numerous trend studies on ozone were published in the past years (e.g. Eckert et al., 2014; Moreira et al., 2015; Steinbrecht et al., 2017; Ball et al., 2018) showing how ozone developed in the course of time. Drift-corrected ozone trends from MIPAS (Michelson Interferometer for Passive Atmospheric Sounding) space-borne observations (July 2002 to April 2012) range from negative (up to $-0.41\,ppm\,decade^{-1}$) in the tropical stratosphere to positive ($+0.55\,ppm\,decade^{-1}$) at southern mid-latitudes (Eckert et al., 2014). A 20-year continuous mapping of the stratospheric ozone layer at the NDACC site Bern could be achieved. A recent trend analysis by Moreira et al. (2015) showed that ozone recovered by about $3\,\%\,decade^{-1}$ at an altitude of $40\,km$ within the time period 1997 to 2015. Steinbrecht et al. (2017) calculated ozone trends for larger number of ground-based NDACC site observations by different techniques such as FTIR (Fourier-Transform-Infrared-Spectrometer), microwave radiometry or lidar. They found positive trends between 35 and $48\,km$ altitude in the tropics as well as in the the 35 to $65°$latitude bands of the Northern and Southern Hemisphere. More specifically, ozone mixing ratios at $42\,km$ increased by 1.5 (tropics) and 2-2.5 (mid-latitudes) $\%\,decade^{-1}$, respectively. Although total column measurements of ozone show that the ozone layer stopped to decline across the globe, there is some evidence from satellite observations that lower stratospheric ozone continued to decline within $60°\,N$ to $60°\,S$ after 1998, resulting in downward trend of stratospheric ozone columns (Ball et al., 2018).

In order to understand detected water vapor trends in the middle atmosphere, models and measurements are both important. A 40-year (1960-1999) model simulation with the coupled chemistry-climate model (CCM) ECHAM resulted in a global mean stratospheric $H_2O$ increase by $0.7\,ppm$ between 1980 and 1999 (Stenke and Grewe, 2005). Trend estimates in lower stratospheric water vapor strongly differentiate between the NOAA (National Oceanic and Atmospheric Administration) FPH observations at Boulder and merged zonal mean satellite measurements as pointed out by Lossow et al. (2018). The differences reach up to $0.5\,ppm\,decade^{-1}$ and change the signs from positive for the in-situ observations to negative for the processed satellite data. But not only the observations do not agree, also extensive trend estimates from simulations show discrepancies for the location of Boulder and the corresponding zonal mean latitude band around $40°\,N$. An intercomparison of ground-

based microwave and satellite linear trends in the lower mesosphere at an altitude of about $53\,\mathrm{km}$ ($0.46\,\mathrm{hPa}$) within different extended periods shows no consistent picture between the different observations. The following stations were considered in the study by Nedoluha et al. (2017): Lauder, Mauna Loa, Table Mountain, Seoul, Bern and Onsala. Satellite retrievals that were integrated in the intercomparison include ACE-FTS (Advanced Composition Explorer - Fourier Transform Spectrometer),

HALOE (Halogen Occultation Experiment), MIPAS (Michelson Interferometer for Passive Atmospheric Sounding), MLS (Microwave Limb Sounder), SCIAMACHY (Scanning Imaging Absorption Spectrometer for Atmospheric Chartography), SMR (Sub-Millimeterwave Radiometer), SOFIE (Solar Occultation For Ice Experiment) and different data subversions of those. At none of the comparison sites a uniform result of only positive or negative trends could be retrieved. This might be related to the problem that the time periods cover different ranges. Regarding Fig. 8 in Nedoluha et al. (2017) the trends at Bern

range from $+16$ to $-5\,\%\,\mathrm{decade}^{-1}$. However, the majority of $H_2O$ time series, including Aura/MLS, exhibit small positive relative trends in the range $1\text{-}7\,\%\,\mathrm{decade}^{-1}$. At the $0.46\,\mathrm{hPa}$ pressure level the multi-linear regression model used in our study does not produce a significant trend at the $95\,\%$ confidence level.

On a seasonal time scale mesospheric water vapor is changing its concentration mainly due to the vertical advection caused by the meridional circulation. As shown by Chandra et al. (1997), within the soloar cycle time scale the modulation of the

Lyman-$\alpha$ radiation intensity is forcing changes up to $30\text{-}40\,\%$ near the mesopause level. An in-situ source of $H_2O$ is the oxidation of methane. The long-term increase in methane accounts thus to an increase in $H_2O$ and estimates yield values about $0.4\,\%$ per year (Chandra et al., 1997). It is clear, that the actual long-term development of mesospheric $H_2O$ is related to a complex mixture of different processes and still it is not certain how mesospheric water vapor develops in a changing climate of the earth. Therefore it is very important to continue the observations especially from those instruments that already have long

records such as the microwave NDACC instruments at Mauna Loa (Hawaii), Table Mountain (USA) or Bern (Switzerland). In this study we report on a detected decline of $H_2O$ in the mesosphere from the NDACC ground-based microwave measurement site Bern in the time period between 2007-2018.

Section 2 introduces the NDACC measurement site Bern with the MIAWARA radiometer in more detail and presents the water vapor data set that is processed in the trend model which is introduced in Sect. 3 later. The final results of the trend study

are handled in Sect. 3.2, while conclusions are given in Sect. 4.

## 2   The MIAWARA radiometer

The MIddle Atmospheric WAter vapor RAdiometer (MIAWARA) measures the intensity of the pressure broadened emission of $H_2O$ molecules at a center frequency of $22.235\,\mathrm{GHz}$ (Kämpfer et al., 2012). Atmospheric pressure decreases exponentially with altitude and this information is reflected in the $H_2O$ line shape. The obtained spectra are used to retrieve water vapor profiles by

means of radiative transfer calculations and the Optimal Estimation Method as described in Rodgers (2000) using the retrieval software package ARTS/qpack (Eriksson et al., 2005; Buehler et al., 2018). As spectroscopic $H_2O$ model a combination of the H2O-MPM93 model from Liebe et al. (1993), for the pressure broadened half line width, and recent entries in the JPL (Jet Propulsory Laboratory) line catalog, for the lower state energy and line strength at $300\,\mathrm{K}$, is taken. MIAWARA is continuously

**Table 1.** MIAWARA technical specifications

| | |
|---|---|
| Calibration | Tipping curve and balancing calibration |
| Operational mode | SSB* 50 dB suppression |
| Line of view | $\sim 20°$ elevation (northward) |
| Mirror | Plane aluminum mirror |
| Antenna | Corrugated horn (HPBW**: 6°) |
| Receiver temperature | $\sim 180$ K |
| Spectrometer | Aqiris FFTS |
| Total bandwidth | 1 GHz |
| Spectral channels | 16385 |

*single sideband | ** half power beamwidth

operated on the roof of the building for Atmospheric Remote Sensing in Zimmerwald (46.88°N, 7.46°E, 907 m a.s.l.), which is close to Bern, since September 2006. The reason why we only use data since April 2007 is a major upgrade of the instrument from optoacoustic to Fast Fourier Transform (FFT) spectrometry. In the course of this upgrade the spectral resolution increased from 600 to 61 kHz. Other technical instrumental parameters are summarized in Table 1.

In the last years, data from the MIAWARA radiometer was used to detect a solar induced variability of mesospheric $H_2O$ (Lainer et al., 2016), further it was used to investigate planetary 16-day, sub-diurnal and 2-day atmospheric wave activities by using $H_2O$ as a dynamical tracer (Scheiben et al., 2014; Lainer et al., 2017, 2018).

## 2.1 Measurement stability

The total FFT spectrometer bandwidth of MIAWARA is 1 GHz, but only a narrow part of maximal 250 MHz is in general
usable in the retrieval procedure due to baseline artifacts at the wings of the $H_2O$ line spectrum. However, the reduced bandwidth is sufficient for the retrieval of water vapor in the middle atmosphere and even less is needed for the mesosphere. In order to guarantee a high stability of the spectral measurements we further constrain the bandwidth to 80 MHz around the central line frequency of MIAWARA. The calibration of the radiometer is done via a tipping curve scheme, using different sky elevation angles, to derive tropospheric opacities and receiver temperatures every 20 minutes (Fig. 1). At several times per
year a manual liquid nitrogen calibration is performed as verification method. Figure 1 demonstrates, that the tipping curve calibration performed well during the whole investigated time period. The seasonal changes in tropospheric opacity are due to the local weather variability and affect the sensitive altitude region of the water vapor retrieval. In order to reduce the effect of tropospheric conditions on the retrieval, we use a variable integration scheme of the spectral information to reach a stable measurement noise of $0.01 \pm 0.0005$ K). Further, we set the measurement response to 80 % to derive a quite stable upper and
lower limit of the measurements. This approach generates profiles with a time resolution of typically a few hours in winter and up to 1-2 days during summer.

The change of a broken pre-amplifier in the MIAWARA frontend in early 2014 resulted in a continously increase of the receiver temperature afterwards. As shown in the bottom plot of Fig. 1, the receiver temperatures were at a rather constant level below $150 \, \mathrm{K}$ before the amplifier change, while thereafter an increase up to about $200 \, \mathrm{K}$ until 2018 has been observed. However this increase does obviously not effect the derivation of the tropospheric opacities which do not show any pattern change or increase after 2014. The increasing receiver temperatures lead also to higher noise levels of MIAWARA. But with the application of a dynamic integration scheme this effect is fully compensated.

The a priori water vapor information is derived from a monthly mean zonal mean climatology using Aura/MLS v2.2 data over 4 years between 2004 and 2008. The most recent Level2 Aura/MLS data (v.4.2) are used to initialize pressure, temperature and geopotential height within the MIAWARA $H_2O$ retrieval. The vertical resolution of the instrument varies between $11 \, \mathrm{km}$ in the stratosphere and $14 \, \mathrm{km}$ in the mesosphere (Deuber et al., 2005). An instrument validation against Aura/MLS v3.3 with more than 1000 seasonal separated profile comparisons can be found in Lainer et al. (2015). An area of $800 \times 400 \, \mathrm{km}$ (E/W $\times$ N/S) has been used as spatial coincident criterion for the satellite overpasses. In the pressure range of $2\text{-}10 \, \mathrm{hPa}$ the relative differences are below $3 \, \%$ and between $0.05\text{-}2 \, \mathrm{hPa}$ the analysis revealed negative biases of MIAWARA compared to Aura/MLS of up to $-10 \, \%$.

With Fig. 2 we show the overall yearly statistics of the MIAWARA residuals in a bandwidth of $80 \, \mathrm{MHz}$. The residuals are defined as the difference between the observed difference spectrum and the modeled spectrum from the retrieved profile and is illustrated as residuum brightness temperature fluctuations $T_R$. Especially measurements at lower altitudes like in the stratosphere are particularly dependent on a good baseline fitting over a broad frequency range. Overall two differnt baseline fittings are performed. A polynomial fit of fifth order and a sinus fit with 6 coefficients guarantee a stable removal of baseline artifacts on our calibrated spectra. In particular, the histograms show the PDF (probability density function) of the binned (bin width: $5 \cdot 10^{-3} \, \mathrm{K}$) brightness temperature fluctuations $T_R$ of the yearly cumulated MIAWRARA measurement noise together with the fit of a normal distribution. Overall, the maxima of the normal distribution fits are centered at $0 \, \mathrm{K}$ and the changes between the years are negligable.

The two plots in Fig. 3 show the monthly and yearly averaged time series of $T_R$ at $22.235 \, \mathrm{GHz}$ valid for the time period between April 2007 and May 2018. In the monthly mean overview it is visible, that the range of the noise varies between $0.0102$ and $0.0097 \, \mathrm{K}$. Starting from autumn 2010 an improvement of the residual temperature patterns could be achieved according to an upgrade of the measurement cycle scheme resulting in more measurement data per time interval, while maintaining the same thermal noise level of the measured difference spectrum. The upgrade of the measurement cycle had no effect on the overall homogeneity of the water vapor time series, also because the measurements were always conducted with the same FFT spectrometer. In both plots no trend pattern can be found, concluding that no frequency shift of MIAWARA occurred within the investigated time period.

Beside baseline artifacts which are not fitted correctly, it is known that the retrieval averaging kernels $\mathbf{A}$ can have an impact on the $H_2O$ profile product. For a long-term measurement-based trend study it is of importance that any variability of $\mathbf{A}$ does not imply a data drift, which could induce an artificial trend. Accordingly we investigate this issue by a sensitivity trend test in Section 3.1.

## 2.2 H₂O data and error handling

Figure 4 presents the derived monthly mean $H_2O$ data time series from the MIAWARA instrument at the northern mid-latitude observation site Bern. From 2007-04-01 to 2018-04-30 a total of 133 months are available. The white horizontal lines indicate the pressure level where the measurement response (MR) drops below $80\%$. A not significant variability of MR can be seen at the lower altitude limit at around $3\,hPa$. A larger but stable variability can be found in the upper mesosphere between 0.02 and $0.04\,hPa$. We find a high correlation between the variability of tropospheric opacity (Fig. 1) and the MR at the upper altitude limit. That the MR variability is not critical for trend estimates is explained in Sec. 3.1.

The annual cycle of water vapor is the most obvious signature in Fig. 4 and mainly originates from dynamics. In the summer mid-latitude mesosphere an upwelling motion of water vapor rich air, caused by the Brewer-Dobson circulation, determines the seasonal variability. The photodissociation by Lyman-$\alpha$ radiation which is stronger during summer has only a minor impact on the abundance of water vapor. This is predominantly the case in the upper mesosphere and mesopause region at about $80\,km$.

For the trend model it is very important to assess a reasonable uncertainty of the microwave radiometer measurements and thus the overall error of the monthly mean water vapor profiles. Two different types of errors were considered. The first type is the natural variability, which can be approximated by the standard error $\sigma_{std}$ of the monthly mean $H_2O$ profiles. The second type is the instrument related observational error $\sigma_{obs}$ that belongs to the random error and depends on the thermal noise on the water vapor spectra. The observational error is calculated during the retrieval computation. Both errors were then combined in the following way to get a total monthly mean error profile $\sigma_{tot}$ for the initialization of the trend model:

$$\sigma_{tot} = \sqrt{\sigma_{std}^2 + \sigma_{obs}^2} \tag{1}$$

The third panel (c) of Fig. 5 shows the temporal evolution of the total error at an altitude of $70\,km$. At this altitude the error predominantly fluctuates around $0.3\,ppm$.

## 3 Trend model description

We performed the trend analyses of the water vapor data through a robust multilinear parametric trend estimation method developed by von Clarmann et al. (2010). The trend program finds a linear trend of the data time series by minimizing a cost function.

The cost function includes a quadratic norm of the residual between a regression model and the analyzed monthly $H_2O$ profile time series, weighted by the inverse covariance matrix of the data errors. The data errors are based on the monthly standard deviation and observational errors of the instruments as described in Sect. 2.2. In addition, error correlations between data points are supported which makes the method suitable for consideration of auto-correlated residuals. The regression

function $Y(t)$ itself consists of an axis intercept, a linear trend, sine waves, and different proxies:

$$Y(t) = a + b \cdot t + c_1 \cdot qbo_1(t) + d_1 \cdot qbo_2(t) \tag{2}$$
$$+ e \cdot F_{10.7}(t) + f \cdot MEI(t)$$
$$+ \sum_{n=2}^{m=3} \left[ c_n \cdot sin \left( \frac{2\pi \cdot t}{l_n} \right) + d_n \cdot cos \left( \frac{2\pi \cdot t}{l_n} \right) \right]$$

where t represents the time, a and b the constant term and the slope of the fit. The terms $qbo_1$ and $qbo_2$ are the normalized Singapore winds at 30 and $50\,hPa$ pressure levels as provided by the Free University of Berlin via http://www.geo.fu-berlin.de/met/ag/strat/produkte/qbo/index.html. According to Kyrölä et al. (2010), the Singapore zonal wind series at the two altitudes are in good approximation orthogonal to each other so that the combination of both can reproduce the Quasi-Biennial Oscillation (QBO) phase shift. Fitting against the solar irradiance variability is accounted for by the $F_{10.7}$ flux which is a good proxy

for this variability. The $MEI$ term in the regression function is the Multivariate ENSO index. It describes the strength of the El Niño - Southern Oscillation (ENSO) with six parameters consisting of surface winds (zonal and meridional), sea surface temperature, sea level pressure, surface air temperature and the sky cloudiness fraction. Both, the solar activity and $MEI$ index lists are available from the following web page: www.esrl.noaa.gov/psd/data/climateindices/list.

The sum term consists of two sine and cosine functions with the period length $l_n$, including the annual and semi-annual

oscillations ($l_1 = 182.5\,d$ and $l_2 = 365\,d$). All coefficients ($a$, $b$, $c_1$, $c_2$, $c_3$, $d_1$, $d_2$, $d_3$, $e$ and $f$) are fitted against the water vapor monthly mean time series in order to estimate the linear variations.

For the water vapor trend analyses, the multi-linear regression model needs the monthly mean profiles together with their uncertainties as input. Figure 5a represents the $H_2O$ model fit (magenta line) on top of the monthly mean time series (blue line) derived by MIAWARA and the linear variation (black line) on $0.04\,hPa$. Overall, the temporal $H_2O$ variability could be

very well reproduced by the model fit, which is also revealed by the residual between the measurements and fit (Fig. 5b) rarely exceeding $0.5\,ppm$. Overall, the regression model is able to explain about $90\,\%$ of the variance of the measurements between $0.02$ and $3\,hPa$. The three other panels display the $H_2O$ fitted signals of the QBO (green line), solar F10.7 cm flux (red line) and ENSO (cyan line) proxies at $0.04\,hPa$ ($70\,km$).

### 3.1 Averaging kernel sensitivity test

Here we describe a performed test on an artificial water vapor profile time series in order to check if the variability of the MIAWARA averaging kernels can induce a data drift that might be misinterpreted as a trend. The averaging kernel matrix **A** is defined as

$$\mathbf{A} = \frac{\partial \hat{x}}{\partial x} = \frac{\partial \hat{x}}{\partial y} \frac{\partial y}{\partial x}. \tag{3}$$

It represents the sensitivity of the retrieved state $\hat{x}$ to the difference in the true atmospheric state $x$. The measured microwave

spectrum is denoted as $y$. In our case we use a time series of one constant artificial $H_2O$ profile $x_{art}$ of $5\,ppm$ at 50 pressure

levels between 10 and $0.01\,\mathrm{hPa}$ at the same time steps as the original MIAWARA profiles were

$$\hat{x}_{art} = x_a + \mathbf{A} \cdot (\mathrm{x}_{\mathrm{art}} - \mathrm{x_a}). \tag{4}$$

$\mathbf{A}$ has to be given on the grid of $x_a$ and is interpolated to the grid of $x$, conserving the measurement response. The artificial convolved water vapor time series $\hat{x}_{art}$ (2007-04 to 2018-04) was then used to calculate monthly mean profiles that could be

used as input to the trend model described in Section 3. No significant trend has been generated by the convolution process with the MIAWARA v301 averaging kernels, the retrieval version for the main trend analysis. In conclusion this means that neither a variability of $\mathbf{A}$ nor a variability in the measurement response (white lines in Fig. 4), which is derived from $\mathbf{A}$, can have an effect on the result of the trend estimate presented in Section 3.2.

## 3.2   H$_2$O trend estimate

After having shown that MIAWARA is measuring with a high instrumental stability, we are confident to present the trend result from the multi-linear parametric trend model (von Clarmann et al., 2010). Figure 6 shows the estimated water vapor trend profiles in absolute (left) and relative (right) values. The latter is calculated relative to the mean H$_2$O profile between April 2007 and May 2018. Although the pressure range of the trend profile goes from $0.01$ to $10\,\mathrm{hPa}$ in the two plots, equivalent to 30-80 km, we restrict the trustworthy trend results to the altitudes of the MIAWARA radiometer which are to a degree of $80\,\%$

a priori independent. These lower and upper limits are marked by the horizontal red lines and are located at $0.03$ and $2.5\,\mathrm{hPa}$. At higher and lower altitudes the trend turns towards zero which is to be expected due to the fact that the MIAWARA mixing ratios gradually approach the climatology of Aura/MLS a priori values and those exhibit no long-term variability. Further not at every pressure level between the red lines a significant trend result could be obtained. This circumstance is expressed by the dashed green boxes by encompassing two altitude regions where the trend is two times larger than the uncertainty. According

to Tiao et al. (1990) this is equivalent to a significance on the $95\,\%$ confidence level.

Below the stratopause from 1 to $2.5\,\mathrm{hPa}$ (42-48 km) a small but still significant negative trend, maximizing at $2\,\mathrm{hPa}$ could be determined. A mean linear decline rate of $-2.5 \cdot 10^{-3}\,\mathrm{ppm\,month^{-1}}$ results in $-0.3 \pm 0.1\,\mathrm{ppm\,decade^{-1}}$ (in relative units: $-4 \pm 1.2\,\%\,\mathrm{decade^{-1}}$) or a total loss of $\approx 0.33\,\mathrm{ppm}$ in the analyzed measurement period. This result is contradictory to explanations presented in North et al. (2015), where the increase of methane in the last decades is expected to also increase the water

vapor content in the stratosphere by photodissociation and oxidation. On the other hand it has been pointed out, that the current understanding of the total stratospheric water vapor budget and the involved mechanisms controlling the entry and mixing of H$_2$O into the lower stratosphere are still under investigation.

The second statistically significant pressure layer in the MIAWARA trend profile is located in the mesosphere between $0.03$ and $0.15\,\mathrm{hPa}$ (61-72 km). Although the $1\sigma$ error in the trend estimate is roughly doubled, the negative trend is clearly

strengthened to $-0.6 \pm 0.2\,\mathrm{ppm\,decade^{-1}}$ at 0.03-0.04 hPa. In relative terms, we see a decrease between $-12$ to $-12.5 \pm 3\,\%\,\mathrm{decade^{-1}}$. The impact of the included extra month of H$_2$O data on the trend estimate was found to be below a change of $\pm 0.05\,\mathrm{ppm}$. It is difficult to find other water vapor trend studies in the literature that investigate mesospheric altitudes and cover a comparable time period. Satellite data from Aura/MLS, which exist since August 2004, could be a basis for trend

investigations. Lately MLS data has been globally analyzed by Froidevaux et al. (2018) and in case of water vapor a positive trend was derived between $100$ and $0.03\,\mathrm{hPa}$ for northern and southern latitudes up to 60 degree. However, Aura/MLS $H_2O$ data below $20\,\mathrm{hPa}$ could be problematic for estimating trends due to detected data drifts (Hurst et al., 2016).

## 4   Conclusions

Robust measurements by the water vapor radiometer MIAWARA, which belongs to the NDACC network, were performed between April 2007 and May 2018 and used to obtain a middle atmospheric trend profile by means of a multi-linear parametric regression trend model fit of prior derived monthly mean profile and uncertainty data time series.

With this study, we demonstrated the high stability of the MIAWARA residuals and outlined that any variability of the averaging kernels or measurement response fluctuations do not induce a measurement drift. Hence we rely on the computed

trend results with the presented multi-linear parametric regression trend model. Overall two altitude regions exhibit a significant ($95\,\%$ confidence) negative water vapor trend during the time period of April 2007 to May 2018:

- $0.03$-$0.15\,\mathrm{hPa}$ ($61$-$72\,\mathrm{km}$): $-12$ to $-12.5 \pm 3\,\%\,\mathrm{decade}^{-1}$

- $1$-$2.5\,\mathrm{hPa}$ ($42$-$48\,\mathrm{km}$): $-4 \pm 1.2\,\%\,\mathrm{decade}^{-1}$

We are not able to give an explanation towards the reasons for the detected $H_2O$ decline below the stratopause and in the

mesosphere. The complexity of interactions between dynamics and chemistry is hardly addressable by observations alone. Numerical investigations will be needed to unravel the impacts of the different processes, like long-term development of methane concentrations, temperature trends, $H_2O$ advection within Brewer-Dobson circulation or changes in photo-dissociation rates.

The fact that a lot of inconsistent results are published, regarding the evolution of middle atmospheric water vapor, it will be of great importance to continue with measurements from various ground-based observation sites. Although satellite missions,

like EOS Aura, can provide data for almost the whole globe ($82°\,\mathrm{S}$ to $82°\,\mathrm{N}$), however the maintenance of the long-term stability and lifetime is limited and complicates trend studies.

*Data availability.* Data from the ground-based microwave instrument MIAWARA is publicly available from the NDACC database as monthly files with a diurnal temporal resolution (ftp://ftp.cpc.ncep.noaa.gov/ndacc/station/bern).

*Competing interests.* The authors declare to have no competing interests.

*Acknowledgements.* The presented study is supported by the Swiss National Science Foundation Grant 200020-160048 and MeteoSwiss in the frame of the GAW project "Fundamental GAW parameters measured by microwave radiometry".

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

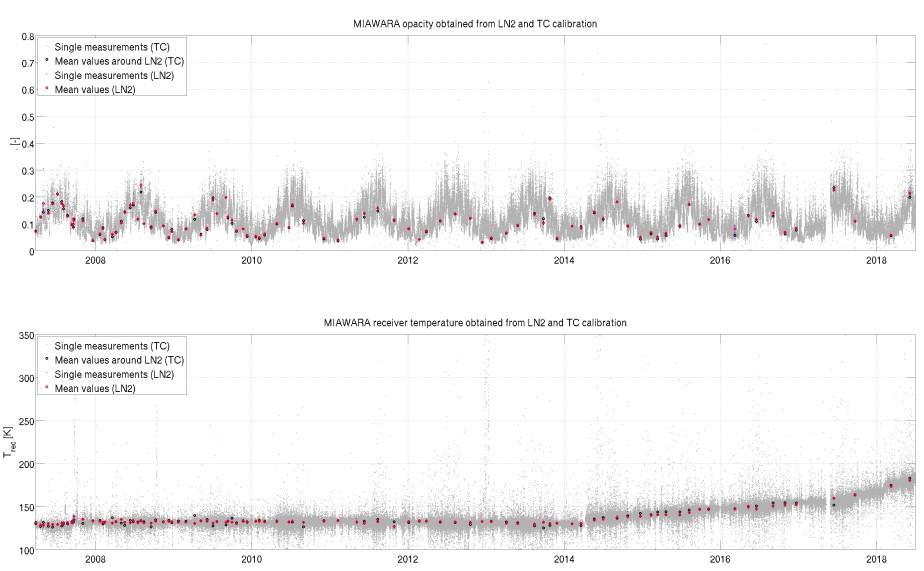

**Figure 1.** Developement of the tropospheric opacities and the MIAWARA receiver temperatures as obtained from tipping curve (TC) (operational,grey dots) and regular liquid nitrogen (LN2) verification calibrations (mean values shown by red markers). The mean values around LN2 can be compared to the mean values around TC that are shown by the black markers. The time period between April 2007 and May 2018 is shown.

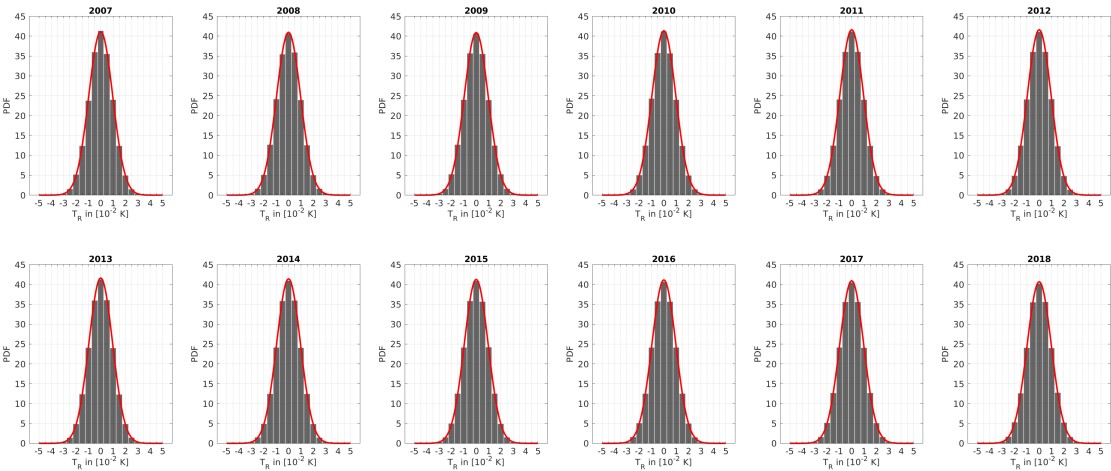

**Figure 2.** Yearly averaged histograms of the MIAWARA residuals (difference between measured difference spectrum and modeled spectrum) as residuum brightness temperature fluctuations $T_R$ in $[10^{-2}\,unit\,K]$ within the frequency range of $22.195\,\text{GHz}$ to $22.275\,\text{GHz}$ ($80\,\text{MHz}$ bandwidth) from 2007 to 2018, showing the evolution of the PDF (probability density function). The red curve is the fit of the corresponding normal distribution. The chosen bin width is $5 \cdot 10^{-3}\,\text{K}$.

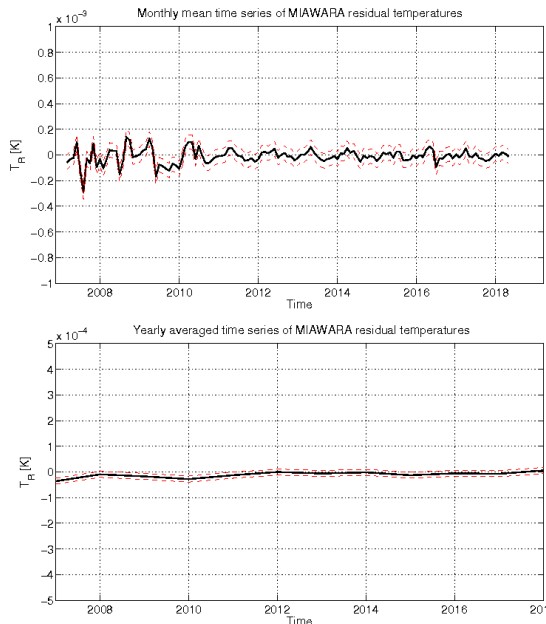

**Figure 3.** Monthly and yearly averaged MIAWARA $T_R$ residuals within the time period of the trend analysis (April 2007 to May 2018). The red dashed lines mark the respective standard deviations.

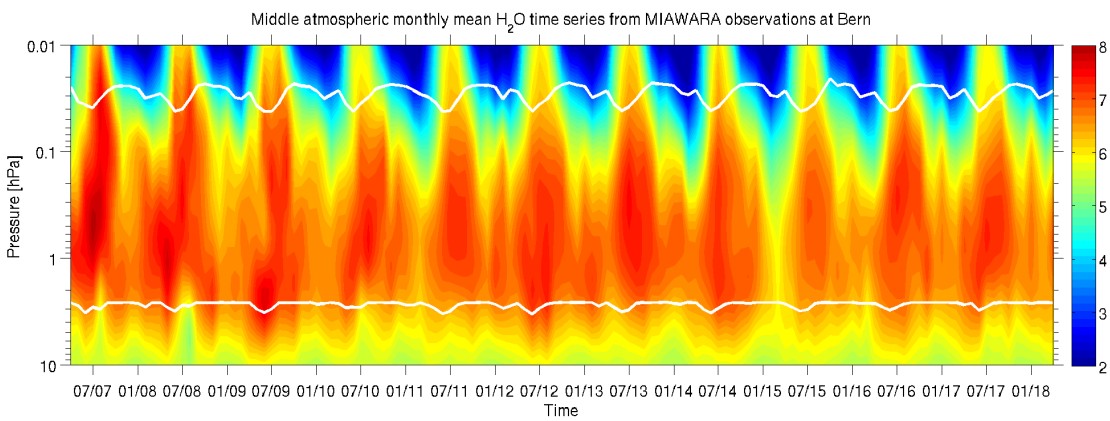

**Figure 4.** Monthly mean water vapor time series in [ppm] obtained by the MIAWARA instrument located at the Zimmerwald observatory near Bern between April 2007 and May 2018. The horizontal upper and lower white lines indicate the pressure layer within which the measurement response is higher than $80\%$. This data set is used as input for the trend model.

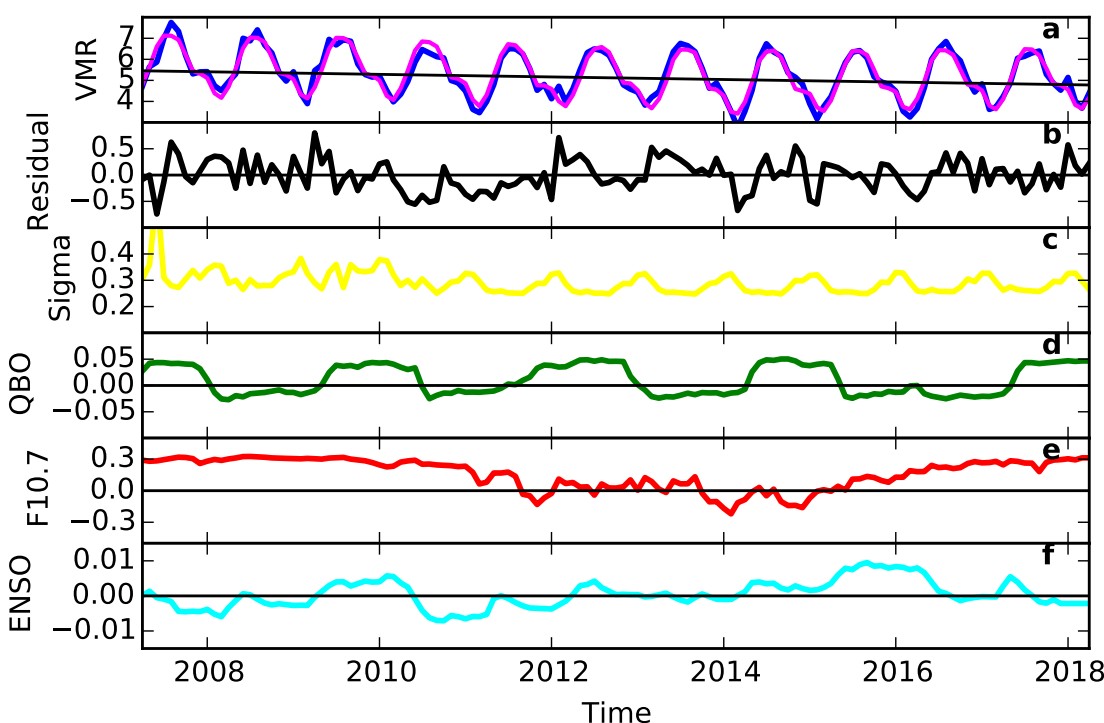

**Figure 5.** Panel (a) shows the trend fit at $0.04\,\mathrm{hPa}$ ($70\,\mathrm{km}$), with the MIAWARA monthly mean $H_2O$ data (blue line), the calculated model fit (magenta line) and the related linear trend (black line). Panel (b) shows the residual and in the following panels (c), (d), (e) and (f) the evolution of the $\sigma$ uncertainty (yellow line), the fitted signals of the QBO (green line), solar F10.7 cm flux (red line) and ENSO (cyan line) proxies at $0.04\,\mathrm{hPa}$.

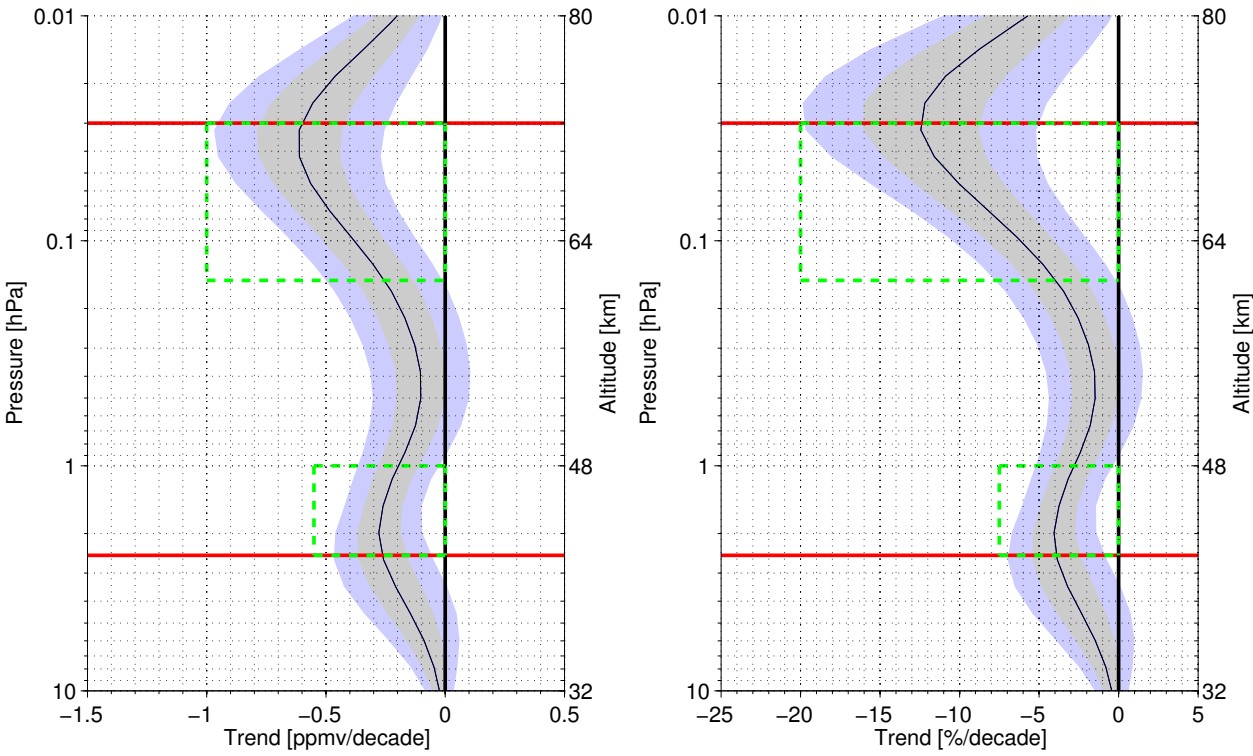

**Figure 6.** Estimated water vapor trend profile in $\left[\mathrm{ppm\,decade^{-1}}\right]$ (left), respectively $\left[\%\,\mathrm{decade^{-1}}\right]$ (rigth), for the time period between April 2007 and May 2018 observed by the MIAWARA instrument at the Zimmerwald observatory close to Bern, Switzerland. The black line represents the trend profile; the grey and violet shaded areas represent the $1\sigma$ and $2\sigma$ uncertainties of the trend estimate. The green boxes show where the trend is statistically significant on the $95\%$ confidence level. The horizontal red lines mark the pressure range (0.03-2.5 hPa) where the MIAWARA data is to $\sim 80\%$ a priori independent.