# Peer review of "Significant decline of mesospheric water vapor at the NDACC site Bern in the period 2007 to 2018"

_Atmospheric Chemistry and Physics, 2018_

## Referee Comment (RC1) · Anonymous Referee #3 · 9 Sep 2018

The work presents 11 years and 1 month of water volume mixing ratio data gathered by a ground-based long-wave instrument operating in Switzerland. The authors fit their retrieved water vapor profiles to a modeled time series so that they can extract a decadal trend of how the water volume mixing ratio has changed.

**1 General point**

I am not convinced that the fitted time model of Equation 2 is good. The figure on this topic, Figure 3, has a yearly variation from 4 ppmv to 8 ppmv in the altitude range the authors selected to show. The residual is about 1 ppmv, up and down to 0.5 ppmv,

or 12-25% of the total volume mixing ratio. This is a lot, especially as the authors find a decadal trend that is of equal or smaller magnitude than the residuals. The authors need to justify these residuals, identify where they are from, and clearly limit the error range of the time model.

**2  Specific points**

**2.1  About Equation 2, the time series**

In Figure 3, the fit seems much more regular over the years than the gathered data. This might be because there are large uncertainties allowed in the fitting mechanism, or because the fit is simply not good. What are the computed uncertainties? Please give error bars in Figure 3.

How are you sure that $F_{10.7}$, the multivariate ENSO index, and the quasi-biennial oscillation phase shift, all only have linear influence on water vapor volume mixing ratios?

What happens to the fit if you switch from monthly to weekly, daily, or a by-the-measurements time series?

Using $c_n/d_n$ and already having defined $c_1$ and $d_1$ is confusing. Also, by your own definitions on page 6 line 24, you never fit semi-annual or annual changes. This does not seem as intended. Can you define $m$, and which $l_n$ you use more precisely? And why limit yourself to just annual and semi-annual trends immediately without decomposing these frequencies from the data first? It is perfectly reasonable to have weather trends that are not exactly annual during such short times as 11 years. And because of the QBO, even lower frequencies seems reasonable to find as well.

Please confirm that the added extra month that makes the time series 11 years and 1 month long has no impact on your results. Its a minor thing, but with such a poor

fit, and with the sharp increase of water vapor there is in Figure 3 around April/May, a single outlier like this can be bothersome.

**2.2 About a priori and retrieval model constraints**

Why the large area for the a priori? You point north, so the southern tip of said area is at your instrument site? Are the coincidences evenly distributed in said area?

You have a 10% difference between your own measurements and those of Aura/MLS. Are these differences constant over the years?

There was a recent conference proceedings paper by Rosenkranz et al [`10.1109/MICRORAD.2018.8430729`] about model errors in the microwave range due to both errors in spectroscopic parameters and the correlation between these errors due to how they are derived in the lab. You never explicitly say so, but I presume you are using his model for the molecular oxygen absorption and possibly even for water in said range, so it seems relevant. If so, the recent paper's findings are important, and they are that there is potential brightness temperature errors of between 0.5 and 1 K in and around the water line you are measuring. What would taking this into account do for your retrievals? Also, please give and cite the spectroscopic model you are using, since this is a user option in ARTS/Qpack.

**2.3 About the water measurements**

Please give specific examples of the fits of Figure 1 for the change that happens around 2011 and explain why you don't believe changing your setup affects the quality of the retrievals from these figures. I can guess you have some sort of standing wave that you can remove in post via periodograms or whatever your favorite deconvolution method might be. I do not think I should be guessing these things though, since it makes the

study less repeatable. So a couple of plots with the measured and fitted line in the center, and an explanation why it is clear that the results are the same both pre- and post-2011 in terms of water vapor would help.

**3  Technical notes**

The entire discussion about ozone in the introduction is irrelevant for the rest of the paper. Please remove it.

Equation 3 should not use $y$ since it is already used in Equation 2. Please change either one of these equations.

Please give all the fitted parameters for Equation 2 in a table or in a figure for different altitudes.

All paper May as such and not Mai.

Page 1 line 15. Please reformulate the first sentence to clarify what is characterizing what and how it is characterizing it. I can guess what you mean but it is unclear.

Page 1 line 21. Please tell for what year the 0.05 W/m$^2$ is from.

Page 3 line 3-4. Please cite and give the full name of each instrument.

Page 7 line 25: according.

---

## Referee Comment (RC2) · Anonymous Referee #2 · 9 Oct 2018

The paper is interesting, important, well written and fulfills the requirements for ACP. Referee: 3 has given valuable comments that I support. I would also like the authors to add a km altitude scale to the right of figures 2 and 4.

---

## Editor Comment (EC1) · F. Khosrawi (Editor) · 10 Oct 2018

Dear authors,

please find below the comments and suggestions for improvements by referee 1:

The study presents a trend analysis of the 10-year data set of middle atmospheric water vapor measured by a ground based microwave radiometer. It is emphasized that the measurement does not show any drifts despite some hardware upgrades and changes in the calibration cycle. Significant trends are found in the mesosphere and upper stratosphere.

The paper addresses an important topic and makes potentially a valuable contribution. However, major revisions are needed to present the necessary evidence, that the measurement does not show significant drifts. The analysis of the baseline is not convincing since it is rather an analysis of the noise instead of the baseline. Changes in the noise level are evident and its consequences on the measurement not sufficiently discussed. The data set does not seem to be homogenized. Despite a dynamic integration scheme to keep the noise level constant there are important variations in the measurement response. The test of the stability of the averaging kernels is over stressed and does not prove that there is no drift in the measurement.

Specific comments:
The Introduction is focused on troposphere and stratosphere but the main results are in the mesosphere. Please adapt the focus of the introduction.

P4/l6: this is not the SNR but the noise.

P4/l14: with 80 MHz bandwidth there is no sensitivity at 10 hPa. Hence the difference simply refers to the difference between MLS climatology (a priori) and MLS measurements. Further, it would be interesting to know, if the bias of 10
P4/l16: Figure 1 does not show anything about the stability of the baseline but only about the evolution of the measurement noise. To demonstrate the stability, please show the annual averages of the residuals instead.

The annual cycle that is visible in Figure 1 contradicts the statement on p4/l6 that the variable integration time ensures a constant noise level. The explanation given on p5/l1 do not apply since the variable integration time should account for all such effects. Further, it is not precise to say the SNR is constant, since if the noise is kept constant with dynamic integration, the line strength can still vary and modify the SNR. The change in pattern after 2016 have to be discussed as well.

P5/l2: I do not agree with the statement that such changes do not affect the retrieval. Changes in the measurement noise affect the sensitivity of the retrieval and can in turn affect the trend analysis.

P5/l14: Why does the white line show such a pronounced seasonal cycle if the measurement error is supposed to be kept constant?

P5/l21: if the observational error is essentially a statistical error, should it not decrease when calculating the monthly mean? Hence why is not $\sigma_{obs}/sqrt(N)$ the correct value?

P5/l29: For dataset shorter than one solar cycle, the solar cycle (SC) term can be highly correlated with the linear term and should be avoided. Are the SC and the linear term correlated in this trend study? What would be the trend results without the SC proxy?

P6/l29: the test with the AVK is much appreciated and a very good indication for the good quality of the data set. However, it does not prove, that there is no shift/steps in the data set which could arise for instance form a drift in frequency. Please comment.

P8/l5: positive drift detected by Hurst et al on MLS are at a pressure levels of greater than 20 hPa. The authors cannot justify the difference between MIAWARA and MLS trends in the low mesosphere by this drift

P8/l11: I do not agree:
- Stability of the baseline has not been shown, see comment above.
- It has been shown, that the actual AVKs do not introduce a drift. It has not been shown that the instrument has no drift. More evidence would be helpful to convince

the reader.

---

## Editor Comment (EC2) · F. Khosrawi (Editor) · 11 Oct 2018

The comment on P4, L14 should correctly read:

P4/l14: with 80 MHz bandwith there is no sensitivity at 10 hPa. Hence the difference simply refers to the difference between MLS climatology (a priori) and MLS measurements. Further, it would be interesting to know, if the bias of 10% towards MLS is constant over time?

---

## Author Comment (AC1)

**Significant decline of mesospheric water vapor at the NDACC site Bern in the period 2007 to 2018**

*Martin Lainer (on behalf of all co-authors)*
* * *
**Response to comments on ACPD paper acp-2018-711**
* * *
**Color Code:** Referee comments, Authors response, Proposed changes in manuscript
* * *
We would like to thank all anonymous referees and the Editor for their constructive comments to our discussion paper.

Please find our point by point responses to the reviewers for the first revision stage below, together with our suggestions to change the manuscript. At the end we included a marked-up version of the updated manuscript.

**1 Response to Referee #1**

The study presents a trend analysis of the 10-year data set of middle atmospheric water vapor measured by a ground based microwave radiometer. It is emphasized that the measurement does not show any drifts despite some hardware upgrades and changes in the calibration cycle. Significant trends are found in the mesosphere and upper stratosphere.
The paper addresses an important topic and makes potentially a valuable contribution. However, major revisions are needed to present the necessary evidence, that the measurement does not show significant drifts. The analysis of the baseline is not convincing since it is rather an analysis of the noise instead of the baseline. Changes in the noise level are evident and its consequences on the measurement not sufficiently discussed. The data set does not seem to be homogenized. Despite a dynamic integration scheme to keep the noise level constant there are important variations in the measurement response. The test of the stability of the averaging kernels is over stressed and does not prove that there is no drift in the measurement.

- The analysis of the baseline as stated in the manuscript is indeed at a first instance an analysis of the measurement noise. However, indirectly we show the good stability of the baseline fitting in the retrieval algorithms. The changes in noise patterns are visible in the 3-dimensional view, but very tiny and would not be recognizable in a 2-dimensional plot looking from above. We do not see any severe changes in noise levels, only small patterns originating either from temperature fluctuations or changes in tropospheric attenuation of the line signal or a combination of both.

- Homogenization would have been necessary if for example a replacement of the spectrometer would have taken place. But did not. Only adjusting the measurement cycle and installing a faster mirror motor does not imply to do a homogenization.

- The periodically variations of the measurement response (Fig. 1) originate from the seasonal variability of the $H_2O$ line strength. We note that these changes in the measurement response do not seem to be important for the trend, because the a priori (MLS) information does not have any trend.

- We think the AVK test is the best way to show the stability of the water vapor measurements. Any important drift of the measurements would be reflected in the AVK development.

The Introduction is focused on troposphere and stratosphere but the main results are in the mesosphere. Please adapt the focus of the introduction.

- We think that some kind of broader introduction to the topic is useful for the reader. Therefor we would like to keep the information on the upper troposphere and stratosphere.

P4/l6: this is not the SNR but the noise.

- Yes this is correct and we will change the expression.

- Page 4, line 6: Corrected expression "signal to noise ratio" by "the measurement noise".

P4/l14: with 80 MHz bandwith there is no sensitivity at 10 hPa. Hence the difference simply refers to the difference between MLS climatology (a priori) and MLS measurements. Further, it would be interesting to know, if the bias of 10% towards MLS is constant over time?

- Regarding the sensitivity at 10 hPa, we are refering to Lainer et al. (2015), where a retrieval version of 225 MHz bandwidth is used. For stability reasons we only use 80 MHz in this trend study. So the difference statement is still correct. Although data down to 10 hPa is shown, we do not make use of it in the results.

- Regarding the bias evolution to MLS over time, we refer to the answer given to a similar comment of referee #3 in Sec. 3.2.2.

P4/l16: Figure 1 does not show anything about the stability of the baseline but only about the evolution of the measurement noise. To demonstrate the stability, please show the annual averages of the residuals instead.

- Thank you very much for this important hint. It is true, we show the evolution of the residuals. But indirectly we show that our baseline fitting works well, otherwise the residual patterns would be not centered at the zero value.

- With the histograms we already show the good stability of the noise, confirming also that the baseline removal works well.

- We will update the manuscript regarding Figure 1 and state that we show the residual development. Fig. 1 shows that there is no drift in the residuals. It also indicates that we correctly removed the baseline.

- We correct all statements involving the baseline. And we clarify that the residuals do not show a drift. This is due to the correct fitting of the baseline and to the stability of the radiometer.

The annual cycle that is visible in Figure 1 contradicts the statement on p4/l6 that the variable integration time ensures a constant noise level. The explanation given on p5/l1 do not apply since the variable integration time should account for all such effects. Further, it is not precise to say the SNR is constant, since if the noise is kept constant with dynamic integration, the line strength can still vary and modify the SNR. The change in pattern after 2016 have to be discussed as well.

- The 3D view of Figure 1 shows very tiny residual patterns that would not be visible if one looks from above (2D view) onto the plot. Also the histograms confirm that the residuals are very constant over the years.

- To clarify the noise issue, we note, that with the variable inegration time we keep the thermal noise of the measured difference spectrum constant at $0.01\,\mathrm{K}$.

- It is true, that we analyse the residuals which are most important to show the goodness of the retrieval.

P5/l2: I do not agree with the statement that such changes do not affect the retrieval. Changes in the measurement noise affect the sensitivity of the retrieval and can in turn affect the trend analysis.

- Sure, the thermal noise affects the sensitivity of the retrieval, but we keep it constant. Figure 1 is showing the residuals, not the noise itself. Those 2 things should be separated. The white lines in Fig. 2 confirm that the noise level is not drifting. There are periodically patterns, but no up/down trends. We are convinced that this does not effect the trend analysis since the trend model includes an annual oscillation and its harmonics.

P5/l14: Why does the white line show such a pronounced seasonal cycle if the measurement error is supposed to be kept constant?

- It is not the measurement error that is kept constant, but the thermal noise. The line strength changes over the year (annual cycle) due to changing tropospheric attenuation. This makes the measurement response or apriori contribution (white line) change periodically with time.

P5/l21: if the observational error is essentially a statistical error, should it not decrease when calculating the monthly mean? Hence why is not $\sigma_{obs}/sqrt(N)$ the correct value?

- We account for the increased sample number within the computation of the standard error $\sigma_{std} = \sigma/sqrt(N)$.

P5/l29: For dataset shorter than one solar cycle, the solar cycle (SC) term can be highly correlated with the linear term and should be avoided. Are the SC and the linear term correlated in this trend study? What would be the trend results without the SC proxy?

- The inclusion of the solar cycle term is essential for the trend model since upper mesospheric water vapour is sensitive to photolysis by the Lyman-alpha radiation of the sun. The uncertainty of the trend estimate inclusive of the solar cycle term is fully considered by the error analysis. Thus, there is no need to switch off the solar cycle term in the trend model since the trend estimate would be of reduced quality if the solar cycle term is not included.

P6/l29: the test with the AVK is much appreciated and a very good indication for the good quality of the data set. However, it does not prove, that there is no shift/steps in the data set which could arise for instance form a drift in frequency. Please comment.

- There are no indications of a drift of the radiometer. Such a drift is avoided by the tipping curve calibration. The frequency-channel relationship is constant since we operate a FFT spectrometer where the spectrum is digitally analysed.

P8/l5: positive drift detected by Hurst et al on MLS are at a pressure levels of greater than 20 hPa. The authors cannot justify the difference between MIAWARA and MLS trends in the low mesosphere by this drift.

- Regarding our manuscript, we do not justify anything by the study of Hurst et al. (2016). Maybe there is mis-understanding. On page 8, lines 9-10, we only state that "However, Aura/MLS $H_2O$ data could be problematic for estimating trends due to detected data drifts (Hurst et al., 2016)."

- We can add the corresponding pressure levels: "below 20 hPa" to this sentence to clarify.

- Page 8, line 10: Added "..below 20 hPa..".

P8/l11: I do not agree:

- Stability of the baseline has not been shown, see comment above.

- It has been shown, that the actual AVKs do not introduce a drift. It has not been shown that the instrument has no drift. More evidence would be helpful to convince the reader.

- We will adapt the text and say that we have shown the evolution and good stability of the measurement noise (only tiny changes, patterns) and with that the functioning of the baseline fitting.

- We are convinced that the AVK is one of the best variables to show the stability of a measurement time series from a radiometer. If the measurements of MIAWARA would drift, the AVK's would also drift. And we have shown that the AVK's are not drifting.

**2 Response to Referee #2**

The paper is interesting, important, well written and fulfills the requirements for ACP. Referee: 3 has given valuable comments that I support. I would also like the authors to add a km altitude scale to the right of figures 2 and 4.

- Sure, it is a good idea to include also a km altitude scale to Fig. 4, because we also give references in km in the text for this figure. However, for Fig. 2 we did not include it because their is no reference in km given in the text. The pressure altitude nomenclatur is typically used for plots covering the middle atmosphere.

- We added km altitude scales to Fig. 4 on the right hand side of the plots.

**3 Response to Referee #3**

**3.1 General point**

I am not convinced that the fitted time model of Equation 2 is good. The figure on this topic, Figure 3, has a yearly variation from 4 ppmv to 8 ppmv in the altitude range the authors selected to show. The residual is about 1 ppmv, up and down to 0.5 ppmv, or 12-25% of the total volume mixing ratio. This is a lot, especially as the authors find a decadal trend that is of equal or smaller magnitude than the residuals. The authors need to justify these residuals, identify where they are from, and clearly limit the error range of the time model.

- We cannot agree to this statement. First, the fitted time model in Equation 2 is a well established method (von Clarmann et al., 2010) and was successfully used in other middle atmospheric trend studies (Moreira et al., 2015) before. The Referee's concern about the residual between measurements and fit is seen from a wrong direction. It is true that locally the residual can reach about 0.5ppm, so maximal 6-12% of the total VMR. However no drift in the residual is present and the mean over the investigated time series is very low (-0.003 ppm). This shows already that the time model fit does a very good job. In our point of view no additional improvements to limit the error range of the time model is neccessary. Regarding ozone residuals shown in Fig. 7 by Moreira et al. (2015), they even reach higher values up to 1ppm, which is still not a problem for retrieving meaningful trends.

- At a first instance, we have not changed the manuscript. See answer above to first comment of Referee.

**3.2 Specific points**

**3.2.1 About Equation 2, the time series**

In Figure 3, the fit seems much more regular over the years than the gathered data. This might be because there are large uncertainties allowed in the fitting mechanism, or because the fit is simply not good. What are the computed uncertainties? Please give error bars in Figure 3.

- That the model fit is more regular and smoother than the data is expected. The fit in our opinion is quite good and represents well the long-term variability of the measurements. Overall the regression model explains about 90 % of the variance between 0.02 and 3 hPa.

- In the manuscript we added the statement that the regression model explains about 90 % of the variance between 0.02 and 3 hPa.

How are you sure that $F_{10.7}$, the multivariate ENSO index, and the quasi-biennial oscillation phase shift, all only have linear influence on water vapor volume mixing ratios?

- We agree that a nonlinear response of the water vapour volume mixing ratios to the solar cycle, ENSO and QBO is possible. However, an investigation of nonlinear effects exceeds the aim of our trend study.

What happens to the fit if you switch from monthly to weekly, daily, or a by-the-measurements time series?

- By now the used trend model only allows to input monthly mean data sets. So this question cannot be answered in a short time. A future program release might allow to check this, but this is defenitly beyond our influence at the moment.

- At the moment and the current setup of our investigation, we are not able to include such an investigation. No changes made.

Using $c_n/d_n$ and already having defined $c_1$ and $d_1$ is confusing. Also, by your own definitions on page 6 line 24, you never fit semi-annual or annual changes. This does not seem as intended. Can you define $m$, and which $l_n$ you use more precisely? And why limit yourself to just annual and semi-annual trends immediately without decomposing these frequencies from the data first? It is perfectly reasonable to have weather trends that are not exactly annual during such short times as 11 years. And because of the QBO, even lower frequencies seems reasonable to find as well.

- Yes, we agree this is a bit confusing and will be clarified in the manuscript. We will define the used $l_n$ and $m$ and correct our definition on page 6, line 24. We fit semi-annual and annual changes, so the sum term goes from n=2 to m=3. The major contribution comes from the semi-annual, respectively annual variation. We agree that other periodicites could be present in the data as well, but since our time model fit in our opinion is already very good, considering other periodicities (expecially shorter than semi-annual) would not impact the trend result a lot and can be neglected.

- An analysis of the dominant frequencies is not necessary since the residuals of our trend model in Fig. 3 are small and do not contain a dominant frequency component.

- Equation 2: We improved the readability of the regression function by changing m to the value 3. Page 6, line 24: Included $c_3$, $c_4$, $d_3$ and $d_4$, the coefficients for the semi-annual and annual variability. We also defined $l_n$ (the period length) more precisely.

Please confirm that the added extra month that makes the time series 11 years and 1 month long has no impact on your results. Its a minor thing, but with such a poor fit, and with the sharp increase of water vapor there is in Figure 3 around April/May, a single outlier like this can be bothersome.

- Regarding this extra month, we want to notice that we do not see a very sharp increase in VMR at the end of Fig. 2. However to be more quantitative, during the analysis and preparation for this study we piecewise increased the water vapor data time series and made the respective trend calculations as time went by. The impact of the included extra month (April 2018) on the trend estimate was found to be very small (changed the trend estimate less than 0.05ppm). We will notice this finding in Section 3.2.

- Page 8, line 6: Here we add the statement, that the additional month does not behave like an outlier. It does not change the trend estimate results.

**3.2.2 About a priori and retrieval model constraints**

Why the large area for the a priori? You point north, so the southern tip of said area is at your instrument site? Are the coincidences evenly distributed in said area?

- We are not completely sure about what this comment is about. We choose a 400x800km area around the groumd-based measurements site where we calculate mean satellite profiles for comparison. Within this area 2 EOS Aura overpasses per day take place. For the a priori we take something different, exactly a monthly mean zonal mean climatology.

You have a 10% difference between your own measurements and those of Aura/MLS. Are these differences constant over the years?

- Within the paper by Nedoluha et al. (2017) compares annual average differences between coincident $H_2O$ measurements and MLS at 0.46hPa for 6 ground-based sites including Bern (MIAWARA). At this altitude MIAWARA data does not behave worse than other ground based data between 2007 and 2014. Usually it is easier to keep ground-based measurements more stable than satellite data. Thus GB data is often used to validate satellite data.

There was a recent conference proceedings paper by Rosenkranz et al (10.1109/MICRO-RAD.2018.8430729) about model errors in the microwave range due to both errors in spectroscopic parameters and the correlation between these errors due to how they are derived in the lab. You never explicitly say so, but I presume you are using his model for the molecular oxygen absorption and possibly even for water in said range, so it seems relevant. If so, the recent paper's findings are important, and they are that there is potential brightness temperature errors of between 0.5 and 1 K in and around the water line you are measuring. What would taking this into account do for your retrievals? Also, please give and cite the spectroscopic model you are using, since this is a user option in ARTS/Qpack.

- As spectroscopic model we use a combination of the H2O-MPM93 model from Liebe et al. (1993) (for the pressure broadened half line width) and recent entries in the JPL (Jet Propulsion Laboratory) line catalog for the lower state energy and line strength at 300 K.

- Page 3, line 22 ff: We give the above information about the spectroscopic $H_2O$ model as a reference in the text: "As spectroscopic $H_2O$ model a combination of the H2O-MPM93 model from Liebe et al. (1993), for the pressure broadened half line width, and recent entries in the JPL (Jet Propulsion Laboratory) line catalog, for the lower state energy and line strength at 300 K, is taken."

**3.2.3 About the water measurements**

Please give specific examples of the fits of Figure 1 for the change that happens around 2011 and explain why you don't believe changing your setup affects the quality of the retrievals from these figures. I can guess you have some sort of standing wave that you can remove in post via periodograms or whatever your favorite deconvolution method might be. I do not think I should be guessing these things though, since it makes the study less repeatable. So a couple of plots with the measured and fitted line in the center, and an explanation why it is clear that the results are the same both pre- and post-2011 in terms of water vapor would help.

- In Figure 1 the residuals shown and have tiny fluctuation patterns. However the thermal noise of the observed spectra is kept constant at $0.01\,\mathrm{K}$. Theoretically changes in the level of thermal noise would indeed affect the retrieval, thus we keep it constant guaranteeing a stable retrieval performance.

- We will give more details on our baseline fitting method. We apply a polynomial fit of fifth order and a sinus fit with 6 coefficients to our calibrated spectrum. The sinus fit is done by an internal MatLab fitting routine.

- The visible change in residual fluctuation patterns after 2011 is only due to the increased sample size of measurements. The actual $T_R$ peak amplitudes pre- and post 2011 are the same.

- Page 4, line 19ff: We added a sentence on the baseline fitting methods applied. "Overall two differnt baseline fittings are performed. A polynomial fit of fifth order and a sinus fit with 6 coefficients guarantee a stable removal of baseline artefacts on our calibrated spectra".

**3.3 Technical notes**

The entire discussion about ozone in the introduction is irrelevant for the rest of the paper. Please remove it.

- This section about ozone could be deleted, but maybe it is still of interest for the reader, because we also describe trend studies that used the same trend model.

Equation 3 should not use $y$ since it is already used in Equation 2. Please change either one of these equations.

- Ok we agree and will change the small $y$ in Equation 2 to a uppercase $Y$.

- Equation 2: Changed $y(t)$ to $Y(t)$.

Please give all the fitted parameters for Equation 2 in a table or in a figure for different altitudes.

- Such a table would be rather confusing and of limited value since the fitted parameters are different at each pressure level.

- All paper May as such and not Mai.

- Page 1 line 15. Please reformulate the first sentence to clarify what is characterizing what and how it is characterizing it. I can guess what you mean but it is unclear.

- Page 1 line 21. Please tell for what year the $0.05\,\mathrm{W\,m^{-2}}$ is from.

- Page 3 line 3-4. Please cite and give the full name of each instrument.

- Page 7 line 25: according.

- Thank you for those comments, which will be considered as far as possible in the revised manuscript.

- Ragarding page 1, line 21: This number is an average between 1999 and 2016, and not for a specific year.

- We agree to give full instrument names, but no citations, since it is already information from another citation Nedoluha et al. (2017).

- Changed Mai to May troughout the manuscript.

- Page 1, line 15: Adjusted sentence for better understanding.

- Page 1, line 21: Now: Globally averaged (1999 to 2016)...

- Page 3, line 3-4: Full names of each instruments are givem.

- Page 7, line 25: Changed to "According"

**References**

Lainer, M., Kämpfer, N., Tschanz, B., Nedoluha, G. E., Ka, S., and Oh, J. J. (2015). Trajectory mapping of middle atmospheric water vapor by a mini network of ndacc instruments. *Atmospheric Chemistry and Physics*, 15(16):9711–9730.

Liebe, H., Hufford, G., and Cotton, M. (1993). Propagation modeling of moist air and suspended water/ice particles at frequencies below 1000 ghz. In *In AGARD, Atmospheric Propagation Effects Through Natural and Man-Made Obscurants for Visible to MM-Wave Radiation 11 p (SEE N94-30495 08-32)*.

Moreira, L., Hocke, K., Eckert, E., von Clarmann, T., and Kämpfer, N. (2015). Trend analysis of the 20-year time series of stratospheric ozone profiles observed by the gromos microwave radiometer at bern. *Atmospheric Chemistry and Physics*, 15(19):10999–11009.

Nedoluha, G. E., Kiefer, M., Lossow, S., Gomez, R. M., Kämpfer, N., Lainer, M., Forkman, P., Christensen, O. M., Oh, J. J., Hartogh, P., Anderson, J., Bramstedt, K., Dinelli, B. M., Garcia-Comas, M., Hervig, M., Murtagh, D., Raspollini, P., Read, W. G., Rosenlof, K., Stiller, G. P., and Walker, K. A. (2017). The sparc water vapor assessment ii: intercomparison of satellite and ground-based microwave measurements. *Atmospheric Chemistry and Physics*, 17(23):14543–14558.

von Clarmann, T., Stiller, G., Grabowski, U., Eckert, E., and Orphal, J. (2010). Technical note: Trend estimation from irregularly sampled, correlated data. *Atmospheric Chemistry and Physics*, 10(14):6737–6747.

---

## Referee Report (RR1)

Review of revised manuscript by Lainer et al. on "Significant decline of mesospheric water vapor at the NDACC site Bern in the period 2007 to 2018"

Neither the response nor the revisions in the manuscript are satisfying. The response is subjective and lacks scientific argumentation in various places (see examples below). The revisions are minimal and do not respond to my main criticism. I recommend to consider acceptance of the manuscript after a major revisions addressing properly the points raised in my first and second review. I came to this recommendation for two reasons: as laid out in the introduction, only few studies exist on mesospheric water vapor trends. Second, I believe the authors are capable to address the open points properly and to provide the necessary sound discussion.

General comments:

The authors present the residuals from the retrievals which show a step and then a periodic pattern. The authors only speculate where the oscillations could come from (p5/l14 The pattern is likely …). Further the speculative explanation is obviously wrong: Neither temperature fluctuations of the absorbers nor tropospheric attenuation could introduce a change in the noise level of the residuals, since the noise level is kept constant with a dynamic integration scheme (p4/l10).

The 80% measurement response contour shows significant variability, which is not a good sign if trends shall be analyzed. Despite my criticism in the first review, this issue is not discussed in the revised paper.

I consider these two points crucial for a trend study and both must be fully addressed and explained by the authors.

Specific comments on the authors response:

"The analysis of the baseline as stated in the manuscript is indeed at first instance an analysis of the measurement noise. However, indirectly we show the good stability of the baseline fitting in the retrieval algorithms.
If you want to discuss the baseline, why do you present results that are at first instance an analysis of the noise and only indirectly show the stability of the baseline? My suggestion to show annually averaged residuals is completely ignored.

The changes in noise patterns are visible in the 3-dimensional view, but very tiny and would not be recognizable in a 2-dimensional plot looking from above.
Are you really telling me, that I should not worry about all the structure in the residuals because I would not recognize them if plotted in 2D? This is an outstanding lack of scientific argumentation. The phrase on p5/l14 can impossibly appear in a scientific publication.

We do not see any severe changes in noise levels, only small patterns originating either from temperature fluctuations or changes in tropospheric attenuation of the line signal or a combination of both."
"not severe" and "only small" is subjective and qualitative and not convincing.

Homogenization would have been necessary if for example a replacement of the spectrometer would have taken place. But did not. Only adjusting the measurement cycle and installing a faster mirror motor does not imply to do a homogenization.

This is again very subjective. The answer whether or not a homogenization is required is given by the data itself. Numerous tests can be found in the literature.

The periodically variations of the measurement response (Fig. 1) originate from the seasonal variability of the H2O line strength. We note that these changes in the measurement response do not seem to be important for the trend, because the a priori (MLS) information does not have any trend.

The variations in measurement response are not addressed in the revised paper. If the line strength was the origin of the variations in measurement response, wouldn't we expect to have a higher measurement response in summer when there is more water vapor, always keeping in mind, that the noise level is kept constant by dynamic integration? But Fig 2 shows the opposite.

We think the AVK test is the best way to show the stability of the water vapor measurements. Any important drift of the measurements would be reflected in the AVK development.

This is lacking scientific argumentation. I acknowledge your expertise in the field, but to simply tell me that you think this is the best way and to claim all kind of measurement drifts would be seen in the AVK does not convince me. The only ingredients of the AVK are the Jacobian and the covariance matrices and all except the Jacobian are constant in time, since you apply a dynamic integration scheme and keep the noise at 0.01 K. What about a time series of the receiver temperature, monthly or annual averages of the residuals (would show frequency shifts), …

We think that some kind of broader introduction to the topic is useful for the reader. Therefor we would like to keep the information on the upper troposphere and stratosphere.

I did not criticize the broadness of the introduction but the fact that a discussion of the mesosphere is missing and I still think I have a point. I am astonished by the extent to which the authors ignore my comments.

---

## Author Response (AR2)

**Response to review of revised manuscript "Significant decline of mesospheric water vapor at the NDACC site Bern in the period 2007 to 2018" by anonymous reviewer #1.**

Martin Lainer (on behalf of all co-authors), 09.04.2019

**Color code for the document:**

Referee comments, Author responses, Relevant links to changes in manuscript

In summary, we performed a moderate revision and considered the suggestion of the reviewer to improve the analysis of the measurement response, receiver temperature and residuals. This led to the inclusion of three new figures in the revised manuscript version. The new figures enhance the confidence that the retrieved water vapor series are adequate for a trend estimation. The figures also show that there is no need to manipulate or modify the observed time series.

In general our changes in the manuscript follow the content of our replies in black.

**General comments:**

The authors present the residuals from the retrievals which show a step and then a periodic pattern. The authors only speculate where the oscillations could come from (p5/l14 The pattern is likely ...). Further the speculative explanation is obviously wrong: Neither temperature fluctuations of the absorbers nor tropospheric attenuation could introduce a change in the noise level of the residuals, since the noise level is kept constant with a dynamic integration scheme (p4/l10).

See our answers to your specific comments on this topic.

The 80% measurement response contour shows significant variability, which is not a good sign if trends shall be analyzed. Despite my criticism in the first review, this issue is not discussed in the revised paper.

The variation in the lower (in altitude) MR is from our point of view not significant. The upper MR development shows a stable seasonal variation between 0.02 and 0.04 hPa. We showed already with the AVK test (convolution with artificial  $H_2O$  time series) that the MIAWARA MR (which is represented in the AVK) does not introduce a trend in water vapor.

I consider these two points crucial for a trend study and both must be fully addressed and explained by the authors.

**Specific comments on the authors response:**

If you want discuss the baseline, why do you present results that are at first instance an analysis of the noise and only indirectly show the stability of the baseline? My suggestion to show annually averaged residuals is completely ignored.

We thought it could be worth to show the whole development of the noise with the frequency dimension, although we do not understand it at the moment, especially the regular patterns the reviewer was worried about. However we will follow the suggestion of the reviewer and now show monthly and annual averaged residuals at the center line frequency of 22.235 GHz instead.

Section 2.1 (Measurement stability), pages 4-6: An analysis of the monthly and annual averaged residuals is included now.

Are you really telling me, that I should not worry about all the structure in the residuals because I would not recognize them if plotted in 2D? This is an outstanding lack of scientific argumentation. The phrase on p5/l14 can impossibly appear in a scientific publication.

Yes, our statement about the 2D plot was truly not very scientific and will be removed. We admit the 3D plot was not a good idea to show as long as we are not able to explain it properly. But the scale of the regular patterns is indeed very small. The noise changes roughly between 0.0105K and 0.0095K, which gives both rounded values of our target noise level of 0.01K. In the monthly mean overview (s. Fig. 1) we see that the range of the noise is even smaller between 0.0102 and 0.0097 K. Starting from autumn 2010, we see an improvement in the noise patterns (residuals are smaller than before), which is related to an upgrade of the measurement cycle (more measurement data per time interval). Here we really do not see things to worry about. To explain a bit further, the dynamic integration is "discretized" by a single time period where MIAWARA obtains a line spectrum. Thus only a close approach to the 0.01K noise level with the dynamic integration scheme is realistic and this we achieve.

Fig. 1: MIAWARA monthly mean time series of residual temperatures between April 2007 and May 2018. The dashed red lines show the standard deviations.

In the revised manuscript we now show the monthly and yearly averaged residual development. The 3D plot will be removed but we keep the histogram statistics plot.

Section 2.1 (Measurement stability) is updated and the 3D plot is removed together with the text passages related to it, while the histogram statistic plots are still kept.

**"not severe" and "only small" is subjective and qualitative and not convincing.**

The subjective statements will be removed and we will focus on concrete values from the newly introduced Figures.

Section 2.1 (Measurement stability): We included 2 new Figures, showing monthly and yearly averaged residuals as well as the receiver temperature and opacity derived from tipping curve and liquid nitrogen calibrations. In the text we refer to the concrete values from the Figures now.

This is again very subjective. The answer whether or not a homogenization is required is given by the data itself. Numerous tests can be found in the literature.

Here we refer to the performed AVK test. The AVKs stay stable over the investigated time period and we do not see the need of further tests here. Further any homogenization of meso-spheric water vapor would be challenging due to the lack of reliable data sets at these altitudes.

The variations in measurement response are not addressed in the revised paper. If the line strength was the origin of the variations in measurement response, wouldn't we expect to have a higher measurement response in summer when there is more water vapor, always keeping in mind, that the noise level is kept constant by dynamic integration? But Fig 2 shows the opposite.

We agree that the line strength of the difference spectrum cannot be the source of the periodic variation of the MR (measurement response) at high altitudes (upper white line). Due to the fact that during summer the tropospheric opacity is higher than in winter (at the mid-latitudinal observation site), the attenuation of the line is higher in summer. We assume that this overcompensates the effect of the increase in mesospheric H2O in summer regarding the measurement response.

Further it can be precluded that the seasonal variation of the MR impacts the linear trend in water vapor. The temporal evolution also shows that the variation in MR is constant. To summarize, the MR variation cannot be due to the retrieval noise since it is constant. It cannot be due to the line strength variation, since an opposite result would be expected. It is difficult to prove where these variations come from. We assume (sorry for being subjective again) that it is related to the tropospheric opacity at the mid-latitude observation site Bern.

Sections 2.1, 2.2 and 3.1 are updated and discuss more the observed development of the measurement response now (see marked up manuscript version).

This is lacking scientific argumentation. I acknowledge your expertise in the field, but to simply tell me that you think this is the best way and to claim all kind of measurement drifts would be seen in the AVK does not convince me. The only ingredients of the AVK are the Jacobian and the covariance matrices and all except the Jacobian are constant in time, since you apply a dynamic integration scheme and keep the noise at 0.01 K. What about a time series of the receiver temperature, monthly or annual averages of the residuals (would show frequency shifts), ...

With the following Fig. 2 we provide a summary of the MIAWARA calibration. There we show the MIAWARA opacities and receiver temperature development over the trend estimation period (April 2007 to May2018). Both parameters are derived from tipping curve calibrations which are periodically compared against liquid nitrogen calibrations. As Fig. 2 shows, the opacities and receiver temperatures from the tipping curve calibration agree well with the ones calculated from liquid nitrogen calibrations. With beginning of early 2014 we observe a steady increase in receiver temperature. The only instrument part which was replaced at that time was a preamplifier in the frontend of MIAWARA. The increasing receiver temperatures lead to higher noise levels, but since we use a dynamic integration scheme we compensate this effect.

Fig. 2: MIAWARA opacities and receiver temperatures obtained from tipping curve and liquid nitrogen calibrations. The time period between April 2007 and May 2018 is shown.

Section 2.1, page 5, lines 3-8: Added a paragraph on the receiver temperature developments and discussion about.

Finally we present yearly averaged residual temperatures (s. Fig. 3) of MIAWARA to follow up the referee's suggestions. It can be seen that the evolution varies around the zero level with no obvious trend. Especially the period after 2014 (when the receiver temperatures started to increase) shows no significant change of the residuals.